# The Effect of a Smart Diaper Sensor on Incontinence-Associated Dermatitis Afflicting a Sedentary Patient with Cognitive Impairment

**DOI:** 10.3390/jcm14072526

**Published:** 2025-04-07

**Authors:** Sol Lee, Jae-Hyung Kim

**Affiliations:** 1Department of Physical Medicine & Rehabilitation, Eulji University Hospital, Daejeon 35233, Republic of Korea; christinelee1026@gmail.com; 2Department of Physical Medicine & Rehabilitation, Chonnam National University Hwasun Hospital & Chonnam National University Medical School, Jeollanamdo 58128, Republic of Korea

**Keywords:** incontinence-associated dermatitis, smart diaper, urinary incontinence, cognitive deficit

## Abstract

**Background:** Urinary incontinence (UI) is common among hospitalized patients and often leads to complications like incontinence-associated dermatitis (IAD). This risk is heightened among patients with cognitive impairment, as delayed diaper changes can worsen skin conditions. Smart diaper sensors provide a promising solution to these challenges. **Methods**: A 76-year-old woman with cognitive impairment and gait disturbance presented with itchy erythema and rashes consistent with IAD. Conventional treatments offered partial relief but did not resolve the symptoms. The MONIT smart diaper sensor was used in combination with antifungal ointment. The sensor, attached externally to the diaper, monitored moisture levels in real time and sent alerts to a smartphone via Bluetooth when a diaper change was needed. **Results:** The MONIT system significantly improved skin condition, as evidenced by reduced scores on the incontinence-associated dermatitis and its severity (IADS) instrument and the perineal assessment tool (PAT). **Conclusions:** The combined use of the MONIT smart diaper sensor and antifungal treatment effectively managed IAD, reducing its severity and preventing complications. Smart diaper sensors represent valuable tools for improving care for sedentary patients with cognitive impairment, offering innovative support for timely interventions.

## 1. Introduction

Incontinence is common among hospitalized patients. The prevalence of urinary incontinence among hospitalized patients has been determined to be 42% [1]. Patients with urinary incontinence experience functional limitations, social isolation, and negative relationships with family members [2]. Using diapers can worsen skin problems. Although they may keep bedding and clothing clean, diapers lead to constant contact between urine or stool and the skin. Over time, the skin breaks down. Incontinence problems can lead to skin infections. Incontinence-associated dermatitis (IAD) may develop when the skin remains in prolonged contact with stool or urine. This condition is frequently observed among patients in long-term care settings or nursing homes, with reported prevalence rates ranging from 5.6% to as high as 50% [3]. In long-term care facilities, particularly in cases with functional limitations or cognitive impairment, the use of diapers is often unavoidable, regardless of the presence of incontinence. Appropriate diaper changes often do not occur at the proper time due to immobility and cognitive impairment, leading to a high incidence of IAD. IAD is characterized by erythematous skin lesions and inflammation with or without skin swelling. It affects areas larger than the perineum, such as the genitalia, the inguinal area, the buttocks, thighs, and the upper abdomen. It results from a combination of factors, such as excessive moistness caused by urinary and/or fecal incontinence, friction, pH changes, and colonization by microorganisms. To prevent IAD, special care must be taken to keep the skin dry and clean. Although a smart diaper sensor can detect and notify people of wetness, it is not sensitive when used as an indicator of the need for a diaper change for people with severe dementia [4]. However, previous studies have mainly focused on pediatric or general elderly populations. There is limited evidence regarding smart diaper sensor use among adults with cognitive impairment. In particular, there is a lack of analyses comparing existing sensors like Pixie Scientific or DFree with the MONIT system (MONIT Corp., Berkeley, CA, USA), which is designed for real-time notification and multi-user access. Thus, in this study, we aimed to assess the impact of a smart diaper sensor on incontinence-associated dermatitis (IAD) and determine its applicability to individuals with cognitive impairment and functional limitations.

## 2. Case Report

A 76-year-old patient was referred to our department of rehabilitation medicine because of weakness in both lower limbs and gait disturbance. The motor power of both lower limbs was grade 2 according to the Medical Research Council. The modified Barthel Index (Korean version) (K-MBI) score was 34/100, indicating maximal assistance was required for the activities of daily living (ADL) because the patient could not stand or ambulate. Electrodiagnostic findings revealed lumbosacral radiculopathy and multiple lumbosacral stenoses. The Mini-Mental State Examination (MMSE) revealed a score of 12 out of 30, indicating significant impairments primarily in terms of time and place as well as delayed recall. The patient’s global deterioration scale (GDS) score was 5, indicating moderately severe cognitive decline. Brain MRI and MRA showed small-vessel ischemic changes in both periventricular white matter and moderate stenosis of the left proximal internal carotid artery. A diagnostic impression of early-stage neurodegenerative dementia, such as Alzheimer’s disease, was made. The patient was placed on a combined pharmacological management regimen consisting of donepezil and choline alfoscerate. However, despite this medication, there was no remarkable improvement in her cognitive function. The patient required maximal assistance for bladder and bowel control. Furthermore, because of cognitive impairment associated with dementia, the patient was unable to express a voiding sense. During rehabilitation, the patient developed itchy erythema due to diaper use. The patient showed an erythematous lesion rash on her lower back and buttocks (Figure 1A). The patient provided informed consent for both clinical and research purposes.

The incontinence-associated dermatitis and its severity (IADS) instrument and the perineal assessment tool (PAT) were used to assess the patient’s skin condition both before and after her use of a smart diaper sensor system. Studies have reported that perineal skin damage related to incontinence affects approximately 29% to 36% of patients in critical care settings [5,6]. Commonly used incontinence products, such as underpads and adult diapers, have been associated with an increased risk of perineal skin damage due to their tendency to retain moisture, which prolongs skin exposure. The IADS instrument was developed to assess IAD [7]. This tool requires a nurse to evaluate skin damage using a five-point Likert scale, with each of 13 body areas (the perianal skin, the genitalia, the crease between the buttocks, the right lower buttock, the left lower buttock, the right upper buttock, the left upper buttock, the lower abdomen/suprapubic area, the crease between the genitalia and the thigh, the right inner thigh, the left inner thigh, the right posterior thigh, and the left posterior thigh) assigned a score based on the most severe feature observed at that site. A score of 1 was given to sites presenting with pink discoloration. A score of 2 was assigned to anatomical sites presenting erythema in the absence of a rash or epidermal disruption. Sites exhibiting clinical signs consistent with a fungal rash were allocated a score of 3, whereas any degree of skin breakdown was rated as 4. The scoring algorithm ranged from 0 to 52, with 0 indicating the complete absence of erythema, a rash, or skin damage. Lower total scores were indicative of more favorable skin integrity. To address the problem of perineal skin damage, the PAT was developed through a literature review [8]. The perineal assessment tool (PAT) is composed of four subscales, each representing a key factor associated with perineal skin breakdown: intensity and type of irritant, duration of irritant exposure, condition of the perineal skin, and contributing factors related to diarrhea. These subscales were developed to reflect varying levels of risk, with each item scored on a three-point scale ranging from 1 (minimal risk) to 3 (high risk). Each score is accompanied by a descriptor and a detailed explanation corresponding to the severity level. The cumulative score spans from 4, indicating the lowest overall risk, to 12, representing the highest potential risk for skin damage. At baseline, the IADS instrument score was 4, and the perineal assessment tool score was 5. The patient was started on desonide lotion (Desowen^®^ lotion, Galderma, Zug, Switzerland) and antifungal ointment (amorolfine) treatment. The patient still complained of itchy erythematous papules and vesicles in her buttock and inguinal areas. After one week of conventional treatment, the IADS score increased to 18 and the PAT score to 8, indicating a worsening of skin integrity (Figure 1B).

The transition from conventional treatment to MONIT smart diaper sensor (MONIT Corp., Berkeley, CA, USA) use occurred on day 8 of treatment, after a 2-day observation period. The patient started using the MONIT system, which is a smart diaper sensor. The smart diaper sensor is equipped with MONIT learning algorithms that send notifications to a smartphone to alert the user when moisture is detected. The MONIT sensor uses capacitive sensing and Bluetooth Low Energy (BLE) with a signal range of up to 10 m. It has a moisture detection sensitivity of approximately 0.1 mL and a response latency of less than 5 s. The MONIT sensor synchronizes all information through the digital application in real-time. Up to five users can access the information and receive notifications simultaneously (Figure 2).

Originally, it was developed for application to babies. We attempted to apply it to a sedentary patient with IAD for skin care. The MONIT smart diaper sensor was attached to the surface of the diaper. When an alarm sounded on a cell phone connected via Bluetooth, the diaper was changed. The day after introducing MONIT, the patient’s skin symptoms showed notable improvement with the continued use of antifungal ointment. Two weeks later, the IADS instrument score was 1, and the PAT score was 4 (Figure 1C). This outcome suggests a rapid recovery compared to traditional management alone. The changes in the IADS instrument and PAT scores before and after use of the smart diaper sensor are shown in Figure 3.

Ethical approval to report this case was obtained from the Eulji Institutional Review Board (Approval Number: EMCIRB21-102). Informed consent was obtained from the patient’s legal guardian.

## 3. Discussion

The International Continence Society defines UI as any involuntary loss of urine [9]. This condition is particularly prevalent among older adults and individuals with limited mobility or neurological impairments [10]. UI often coexists with Alzheimer’s disease [11]. UI is frequently observed among older adults and has been strongly linked to both cognitive decline, such as dementia, and physical vulnerability, including frailty. The association between dementia and urinary incontinence tends to become stronger with increasing severity and progression of cognitive impairment [12]. Patients with advanced dementia might have a limited ability to communicate and may even have difficulty recognizing medical symptoms. Among elderly patients with dementia, UI has been linked to increased caregiver burdens, which may contribute to the decision to place an individual under nursing home care. Furthermore, UI is associated with a heightened risk of complications such as pressure ulcers, falls, fractures, and urinary tract infections, all of which can result in substantial healthcare costs. Urinary incontinence has been linked to greater degrees of cognitive impairment and an increased incidence of behavioral disturbances among patients with dementia [13]. Among patients with UI, there is a need for comprehensive care approaches that address the physical and psychosocial aspects of this condition. 

Diapers absorb urine and/or stool. The use of a diaper is considered for adults and elderly patients with incontinence or severe mobility limitations who are unable to use assistive devices to manage urinary and fecal elimination. According to the literature, the primary complications related to the use of diapers include pain, discomfort, pressure sores, IAD, and worsening urinary incontinence. Diaper use has also been associated with low self-esteem. It also elevates the risk of nosocomial infection [14]. IAD is the main complication of indiscriminate diaper use. The incidence of IAD had been reported to be 36% among critically ill patients [5], 7.6% among patients receiving long-term acute care [15], and 5.5% among nursing home residents with newly developed incontinence [16]. IAD has been identified as an obvious risk factor for pressure sores. It occurs more frequently among patients with pressure sores. It may contribute to the worsening of existing pressure sores [17]. The development of IAD is associated with significant morbidity and impaired quality of life, particularly among elderly patients with cognitive impairments [18].

To prevent IAD, it is advisable to implement a standardized skin care routine that includes gentle cleansing of the perineal area, regular moisturization, and the use of a moisture barrier or skin protectant. Clinicians do not recommend specific treatments, such as antifungal agents, corticosteroid-based topical anti-inflammatory agents, or topical antibiotics, for the routine management of IAD. Antifungal agents should be applied only when a fungal infection is present. Additional strategies, such as the use of absorbent or containment products and/or indwelling devices, may also be considered in specific cases to support both the management and prevention of IAD [19]. However, traditional methods of managing IAD may be challenging to effectively implement for patients with cognitive impairment, who may have difficulty communicating discomfort or adhering to skincare regimens. Timely changes of wet diapers and maintenance of a dry perineum through the use of a smart diaper system could contribute to IAD prevention. The introduction of smart diaper sensors offers a promising solution for overcoming the challenges associated with traditional IAD management. These sensors allow real-time monitoring of moisture levels in diapers, enabling timely interventions, such as diaper changes and the application of skincare products. UI and IAD present significant challenges in the management of patients, particularly those with functional limitations and cognitive impairments. This case report demonstrates that the use of a smart diaper sensor effectively controlled the skin symptoms and signs associated with IAD for an immobilized patient with cognitive impairment.

In our case, the patient presented with lower limb weakness, gait disturbance, and cognitive impairment, requiring maximal assistance with bladder and bowel control. The patient developed itchy erythema and rashes on their lower back and buttocks, indicative of IAD. Traditional management with desonide lotion and antifungal agents provided partial relief but failed to resolve the symptoms entirely. The subsequent use of the MONIT smart diaper sensor coupled with antifungal ointment application significantly improved skin symptoms and signs within a short period. This demonstrates the potential of smart diaper sensors to facilitate timely diaper changes and prevent prolonged exposure to moisture, thereby reducing the risk of IAD. The MONIT smart diaper sensor offers several advantages for the management of UI and its associated complications. Unlike other commercial sensors such as Pixie Scientific (New York, NY, USA) (which uses embedded sensors) or DFree (San Diego, CA, USA) (which uses ultrasound tracking), the MONIT system combines ease of use with app-based alert systems. MONIT’s real-time, multi-user functionality and compatibility with adult patients make it a unique solution. This case highlights the effectiveness of MONIT in improving the skin condition of a sedentary, cognitively impaired adult with IAD. By facilitating timely diaper changes and maintaining dryness, MONIT helped reduce IADS scores from 18 to 1 and PAT scores from 8 to 4 within two weeks.

## 4. Limitations

Despite these promising results, the smart diaper sensor has several limitations. First, there is a risk of losing the diaper sensor owing to its small size. A cell phone sounds the alarm. However, if an alarm is triggered when the distance between the smartphone and the sensor becomes excessive, it may help prevent the loss of the sensor. Second, considering that the sensor may not be firmly attached to some diapers, the design of the sensor should be improved so that it can be applied to all types of diapers to maximize its usability in clinical settings. Smart diaper sensors hold promise as assistive tools in the management of UI and associated complications, such as IAD, for sedentary patients with cognitive impairment.

## 5. Conclusions

A smart diaper sensor is a helpful tool for IAD management, when combined with an appropriate antifungal agent. However, there are two limitations to the application of smart diaper sensors in treating adult sedentary patients with dementia. First, a smart sensory system has a high risk of becoming lost. An alarm function for detecting a disconnection between the main device and the sensor and an additional application that can find the sensor are required. Second, the sensor cannot be attached to some diapers. The design of the sensor should be enhanced so that it can be applied to all types of diapers. Further research is needed to address the existing limitations and maximize the potential benefits of this sensor technology in clinical practice so that we can enhance the quality of care for patients with UI and improve their overall well-being. The use of a smart diaper sensor can revolutionize UI care and reduce the burden of this prevalent and debilitating condition. While it would be premature to conclude that MONIT “revolutionizes UI care”, our findings support its potential as a valuable assistive tool in the comprehensive care of patients with incontinence.

## Figures and Tables

**Figure 1 jcm-14-02526-f001:**
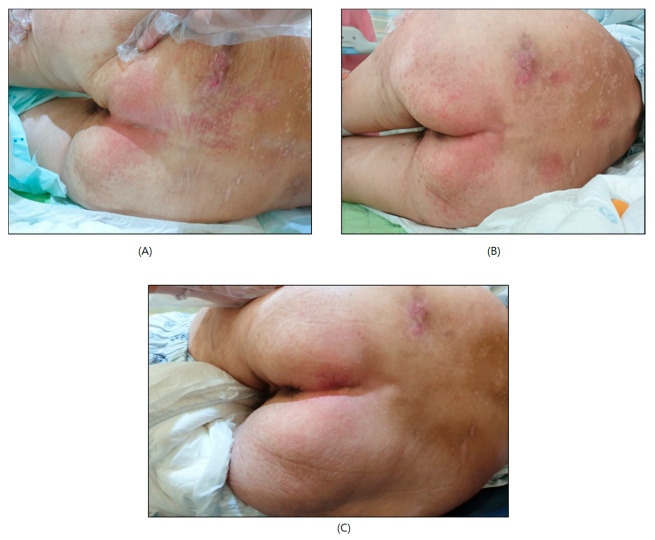
Incontinence-associated dermatitis on lower back and both buttocks: (**A**) initial skin lesion; (**B**) after treatment with only topical agents; and (**C**) two weeks after MONIT application.

**Figure 2 jcm-14-02526-f002:**
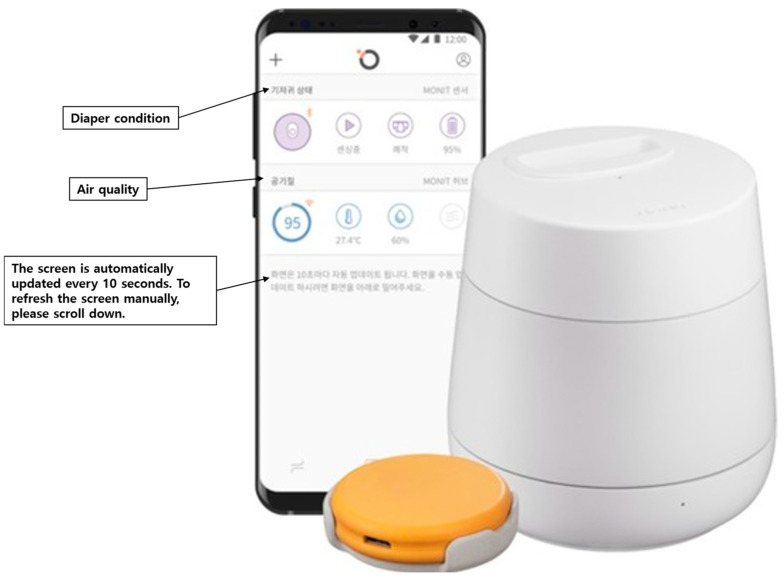
MONIT smart diaper sensor unit.

**Figure 3 jcm-14-02526-f003:**
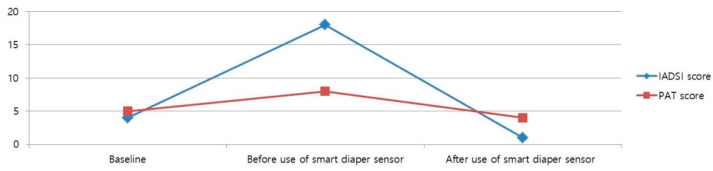
Changes in the perineal assessment tool and incontinence-associated-dermatitis-and-its-severity instrument scores.

## Data Availability

The data from this work will be made available upon request.

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
