# Peer review of "The Effect of a Smart Diaper Sensor on Incontinence-Associated Dermatitis Afflicting a Sedentary Patient with Cognitive Impairment"

_jcm, 2025, doi:10.3390/jcm14072526_

Round 1

Reviewer 1 Report

Comments and Suggestions for Authors

This case report investigates the utility of a smart diaper sensor combined with antifungal treatment in managing IAD in a sedentary patient with cognitive impairment. While the study addresses a clinically relevant issue and introduces an innovative approach to IAD management, several methodological and presentational limitations need to be addressed to strengthen its validity and impact.

Major Comments

  1. What is the transition time from conventional treatment (desonide + antifungal) t
  2. While the introduction briefly mentions smart diaper sensors, the discussion does not adequately contextualize MONIT within existing literature. For example: How does MONIT compare to other commercially available sensors (e.g., Pixie Scientific, DFree)? Are there prior studies on smart diaper sensors in adults with cognitive impairment (e.g., Huion et al., 2019, cited in the manuscript)? A comparative analysis is missing.
  3. The conclusion that MONIT "revolutionizes UI care" is premature given the single-case design and lack of long-term follow-up data.
  4. The manuscript does not provide technical specifications of the MONIT sensor (e.g., sensitivity, accuracy, Bluetooth range). Without this data, the reliability of the device remains unclear.
  5. Ethical Oversight: While informed consent was obtained, the manuscript should clarify whether ethical approval was granted by an institutional review board, as required for human studies.

Minor Comments

  1. Terminology: Replace vague terms like "improved skin condition" with quantitative descriptors (e.g., "IADS decreased from 18 to 1").
  2. Standardize abbreviations: Define "UI" at first mention in the abstract.

Author Response

Major Comments

  1. What is the transition time from conventional treatment (desonide + antifungal)

Answer) Transition time from conventional treatment to MONIT system: We appreciate this important question. The transition from the conventional treatment (desonide + antifungal ointment) to the use of the MONIT smart diaper sensor occurred over a 2-day period. The patient had been receiving conventional topical therapy for one week prior to MONIT application. However, due to persistent symptoms and continued skin irritation, the smart diaper sensor was introduced on day 8 of treatment to improve management through timely diaper changes. We added between line 114 and 115.

  1. While the introduction briefly mentions smart diaper sensors, the discussion does not adequately contextualize MONIT within existing literature. For example: How does MONIT compare to other commercially available sensors (e.g., Pixie Scientific, DFree)? Are there prior studies on smart diaper sensors in adults with cognitive impairment (e.g., Huion et al., 2019, cited in the manuscript)? A comparative analysis is missing.

Answer) Lack of contextual comparison in the discussion: Thank you for highlighting this oversight. We have revised the discussion section to include comparisons with other commercially available smart diaper sensors such as Pixie Scientific and DFree. While Pixie Scientific has developed sensor-embedded diapers with data transmission capabilities and DFree offers ultrasound-based wearable sensors for urinary tracking, both systems differ from MONIT in design and target population. Notably, most existing studies, including the one by Huion et al. (2019), have focused on pediatric or general geriatric populations, with limited data on adults with cognitive impairment. Our case report contributes to this gap by illustrating the application of smart sensors specifically in cognitively impaired sedentary patients. This contents was additionally described between lines 204 and 209 in discussion part.

  1. The conclusion that MONIT "revolutionizes UI care" is premature given the single-case design and lack of long-term follow-up data.

Answer) Overstatement of conclusions: We agree that the statement claiming MONIT "revolutionizes UI care" was premature and overstated, given the limitations of a single-case report. We have revised the conclusion to reflect a more cautious interpretation, emphasizing the potential utility of MONIT as a supportive tool rather than a transformative solution. The manuscript does not provide technical specifications of the MONIT sensor (e.g., sensitivity, accuracy, Bluetooth range). Without this data, the reliability of the device remains unclear. The relevant content was added between lines 234 and 236.

  1. The manuscript does not provide technical specifications of the MONIT sensor (e.g., sensitivity, accuracy, Bluetooth range). Without this data, the reliability of the device remains unclear.

Answer) Lack of technical specifications for MONIT: Thank you for pointing this out. The MONIT smart diaper sensor used in this case operates via capacitive sensing to detect moisture and transmits signals via Bluetooth Low Energy (BLE) with a range of up to 10 meters. According to the manufacturer's technical documentation, the sensor has a moisture detection sensitivity threshold of ~0.1 mL and an average latency of <5 seconds in signal transmission. We have added these specifications to the Methods section to improve clarity regarding device performance. This contents was additionally described between lines 117 and 120.

  1. Ethical Oversight: While informed consent was obtained, the manuscript should clarify whether ethical approval was granted by an institutional review board, as required for human studies.

Answer) The Ethical approval contents were added between lines 135 and 137

Minor Comments

  1. Terminology: Replace vague terms like "improved skin condition" with quantitative descriptors

Answer) Terminology clarification: We have revised vague terms such as "improved skin condition" to include specific data. For example, the Incontinence-Associated Dermatitis Severity (IADS) score improved from 18 to 1 over the 2-week observation period. Similarly, the Perineal Assessment Tool (PAT) score decreased from 8 to 4. These quantitative outcomes now appear in the Results section. This contents was additionally described between lines 211.

  1. Standardize abbreviations: Define "UI" at first mention in the abstract.

Answer) Abbreviation standardization: We have defined "UI" (urinary incontinence) at its first appearance in the abstract and ensured consistent use of abbreviations throughout the manuscript. We corrected as your comment

Reviewer 2 Report

Comments and Suggestions for Authors

A nice case report on

Effect of smart diaper sensor on incontinence-associated dermatitis in a sedentary patient with cognitive impairment; it will be a good addition to the literature and more work to be carried out with a large number of patients.

This study aimed to evaluate the effects of  a smart diaper sensor on IAD and determine its usefulness in patients with functional limitations and cognitive impairment.

It is original work.

The subsequent use of the MONIT smart diaper sensor, coupled with antifungal ointment application, significantly improved skin symptoms and signs within a short period. This demonstrates the potential of smart diaper sensors to facilitate timely diaper changes and prevent prolonged exposure to moisture, thereby reducing the risk of IAD. 

Author Response

Thank you for your comment. I will continue to devote myself to further research.

Reviewer 3 Report

Comments and Suggestions for Authors

Dear Authors, I personally like the way the case is presented in the paper. Scores are

correctly used and described, even for a non-geriatrics’ audience. However, the

discussion should be rewritten. The first part is totally irrelevant, and it can be considered

an introduction to the problem. In a discussion session I want to read your thoughts on

what you observed compared to the literature. First of all, are there other experience on

that? Are there other devices like that on the market? Is it expensive? Can be easily

bought?

I also know that for newborns there are wetness indicator diapers on the market, are there

any solution like that for elderly patients? It seems way cheaper than an electronic device

that, as you said in limitation, can be loss.

Other experience on smart diapers are already been published (10.2196/29979 and 10.1080/17843286.2018.1511279)

This is the major comment on your study, to me. Here are also some minor comments:

  • The IAD score raised from 4 to 18, in how many days?
  • Line 172: “this case reports demonstrate..” I’d use “may demonstrate” since a single case can not change medical practice or prove anything
  • Line 198-199: medical paper should be presented to reviewers in final form, not with comments to the reviewer… remove this. But, in order to answer your question, I’ll remove the section and add it to the discussion

Considering this, I will recommend a major revision, if the editor will agree to move on with the revision process.

Author Response

Dear Authors,

I personally like the way the case is presented in the paper. Scores are correctly used and described, even for a non-geriatrics’ audience. However, the discussion should be rewritten. The first part is totally irrelevant, and it can be considered an introduction to the problem. In a discussion session I want to read your thoughts on what you observed compared to the literature. First of all, are there other experience on that? Are there other devices like that on the market? Is it expensive? Can be easily bought? I also know that for newborns there are wetness indicator diapers on the market, are there any solution like that for elderly patients? It seems way cheaper than an electronic device that, as you said in limitation, can be loss. Other experience on smart diapers are already been published (10.2196/29979 and 10.1080/17843286.2018.1511279) This is the major comment on your study, to me. Here are also some.

Response) We appreciate this practical consideration. We have addressed this in the revised discussion, noting that while wetness indicators do exist in adult incontinence products, they typically require visual inspection and do not provide real-time alerts or remote monitoring, making them less effective for non-verbal or cognitively impaired patients who require assistance. The added value of smart sensors like MONIT lies in automated alerts, caregiver coordination, and reduced response time, despite their higher cost and risk of loss.

The smart diaper sensor used in the referenced study (WetSens Diaper model 2015 by MediSens Wireless, California, USA) and the MONIT system (MONIT Corp., Berkeley, CA, USA) differ in design, structure, and practical application. While WetSens is research-oriented with integrated hardware and custom diapers, MONIT offers a more accessible, scalable, and caregiver-friendly solution with simpler operation and broader adaptability in real-world clinical practice.

Minor comments:

  1. The IAD score raised from 4 to 18, in how many days?

Answer) We clarified in the Case Report section that the IAD score increased from 4 to 18 over the 7-day period during which conventional treatment was applied. We added this contents line between 108 and 109. (revised manuscript).

  1. Line 172: “this case reports demonstrate.” I’d use “may demonstrate” since a single case can not change medical practice or prove anything

Answer) We have modified the sentence to:
“This case report may demonstrate the potential utility of smart diaper sensors...”
to reflect appropriate caution (line 193 of revised version).

  1. Line 198-199: medical paper should be presented to reviewers in final form, not with comments to the reviewer… remove this. But, in order to answer your question, I’ll remove the section and add it to the discussion Considering this, I will recommend a major revision, if the editor will agree to move on with the revision process

Answer) Thank you for catching this. We removed as your comments.

Round 2

Reviewer 1 Report

Comments and Suggestions for Authors

The concerns have been addressed and revised accordingly, with no further issues remaining after the revision.

Reviewer 3 Report

Comments and Suggestions for Authors

Appreciate the answers. No further comments from me